# High-Voltage Cable Buffer Layer Ablation Fault Identification Based on Artificial Intelligence and Frequency Domain Impedance Spectroscopy

**DOI:** 10.3390/s24103067

**Published:** 2024-05-11

**Authors:** Jiajun Liu, Mingchao Ma, Xin Liu, Haokun Xu

**Affiliations:** School of Electrical Engineering, Xi’an University of Technology, Xi’an 710054, China; liujiajun-88@163.com (J.L.); 2211920084@stu.xaut.edu.cn (X.L.); 2221920066@stu.xaut.edu.cn (H.X.)

**Keywords:** buffer layer ablation, artificial intelligence, distributed parameters, fault identification

## Abstract

In recent years, the occurrence of high-voltage cable buffer layer ablation faults has become frequent, posing a serious threat to the safe and stable operation of cables. Failure to promptly detect and address such faults may lead to cable breakdowns, impacting the normal operation of the power system. To overcome the limitations of existing methods for identifying buffer layer ablation faults in high-voltage cables, a method for identifying buffer layer ablation faults based on frequency domain impedance spectroscopy and artificial intelligence is proposed. Firstly, based on the cable distributed parameter model and frequency domain impedance spectroscopy, a mathematical model of the input impedance of a cable containing buffer layer ablation faults is derived. Through a simulation, the input impedance spectroscopy at the first end of the cables under normal conditions, buffer layer ablation, local aging, and inductive faults is performed, enabling the identification of inductive and capacitive faults through a comparative analysis. Secondly, the frequency domain amplitude spectroscopy of the buffer layer ablation and local aging faults are used as datasets and are input into a neural network model for training and validation to identify buffer layer ablation and local aging faults. Finally, using multiple evaluation metrics to assess the neural network model validates the superiority of the MLP neural network in cable fault identification models and experimentally confirms the effectiveness of the proposed method.

## 1. Introduction

Cross-linked polyethylene (XLPE) power cables are extensively utilized in power systems due to their lightweight nature, ease of installation, and outstanding electrical and mechanical properties [1,2,3]. However, throughout construction and operation, cables are susceptible to various external disturbances like mechanical impacts and water ingress, which can compromise their insulation performance, thereby increasing the risk of cable faults [4,5,6]. In recent years, there has been a notable rise in the occurrence of buffer layer ablation faults in high-voltage cables. These faults are highly covert and challenging to detect and prevent. Once they manifest, they may result in cable breakdowns or damage to the metallic sheath. Traditional detection methods struggle to effectively identify and locate such faults, which also exhibit progressive and cumulative characteristics. If left unattended, ablation faults can have severe consequences, posing significant safety hazards to the stable operation of power systems [7,8,9].

The primary methods for detecting buffer layer ablation faults of high-voltage cables include ultrasonic technology detection, X-ray detection, and characteristic gas detection. For example, in reference [10], a method is proposed for detecting cable buffer layer ablation faults using ultrasonic technology. This method involves analyzing the propagation characteristics of ultrasonic waves generated by discharge in the buffer layer and assessing the sensitivity of the detection signal at the cable sheath. While strong discharge occurring in the aluminum sheath valley can be detected using ultrasonic wave detection technology, it becomes challenging to detect discharge signals when they occur in the inner layer of the buffer layer. Another method, as proposed in references [11,12], utilizes characteristic gases for detecting buffer layer ablation faults. By employing gas chromatography-mass spectrometry (GC-MS) technology, the gases produced by cable discharge are analyzed to determine if they contain aromatic hydrocarbon gases, thereby indicating the presence of buffer layer ablation faults. However, this method involves complex detection steps and requires offline testing. Additionally, reference [13] explores the use of deep X-ray digital image processing and intelligent identification technology for power cable detection. This approach achieves the identification of faults in cable buffer layers by combining X-ray digital imaging with intelligent algorithms. However, it is worth noting that this method relies on the presence of locally generated white powder for accurate detection.

Currently, the emerging frequency domain impedance spectroscopy method can reflect the transmission and loss characteristics of high-voltage cables at different frequencies, thereby revealing changes in the internal structure and materials of the cable. There have been studies on frequency domain impedance spectroscopy at the first end of the cable both domestically and internationally [14,15,16]. For example, reference [15] proposes a power cable internal fault identification method based on input impedance spectroscopy, which can identify fault types and locate fault positions based on impedance spectroscopy characteristics. Compared to traditional partial discharge and reflection coefficient spectroscopy methods, this method exhibits a higher sensitivity and accuracy. Reference [16] developed a monitoring scheme based on frequency domain impedance spectroscopy. This scheme detects and locates small voids in the cable insulation through frequency domain impedance measurements, enabling the localization of severe damage to cable insulation and areas of cable deformation. It can be seen that there is some research on the application of frequency domain impedance spectroscopy in identifying and detecting cable aging, local damage, and impedance grounding faults. However, there is relatively little research on applying this method to identify buffer layer ablation faults in cables. Additionally, artificial intelligence has laid a certain foundation in the field of cable fault identification, and compared to traditional cable fault identification methods, it has unique advantages in classification, feature analysis, etc.

Currently, the emerging frequency domain impedance spectroscopy method can effectively capture the transmission and loss characteristics of high-voltage cables across various frequencies, thereby providing insights into changes within the cable’s internal structure and materials. Studies on frequency domain impedance spectroscopy have been conducted both domestically and internationally, particularly focusing on the first end of the cable [14,15,16]. For instance, in reference [15], a method for identifying internal faults in power cables is proposed based on input impedance spectroscopy. This method can identify fault types and pinpoint fault locations by analyzing impedance spectroscopy characteristics. Compared to traditional methods, such as partial discharge and reflection coefficient spectroscopy, this approach offers heightened sensitivity and accuracy. Moreover, reference [16] presents a monitoring scheme leveraging frequency domain impedance spectroscopy. This scheme detects and locates small voids in cable insulation through impedance measurements, facilitating the identification of severe damage and areas of deformation in the cable. While research has explored the utility of frequency domain impedance spectroscopy in identifying cable aging, local damage, and impedance grounding faults, relatively little attention has been directed towards its application in identifying buffer layer ablation faults in the cable. Additionally, artificial intelligence has made significant strides in cable fault identification. Compared to traditional methods, AI offers distinct advantages in classification and feature analysis, laying a solid foundation for further advancements in the field.

For example, reference [17] introduces a groundbreaking time–frequency domain reflection method (TFDR) utilizing an unsupervised neural network model that integrates long short-term memory (LSTM) and variational autoencoder (VAE). This method can effectively detect joint defects and cable faults. In another study, reference [18] presents a method for partial discharge pattern in cable termination based on convolutional neural networks, which can accurately identify various types of defects in cable termination. Additionally, reference [19] successfully identifies local discharge patterns in underground cable joints using convolutional neural networks.

Based on the above analysis, this paper combines the frequency domain impedance method with artificial intelligence to identify buffer layer ablation faults.

The main contributions of this paper are as follows:(1)Utilizing the frequency domain impedance method, impedance spectroscopy is conducted at the first end of the cable to distinguish between inductive and capacitive faults.(2)A method is proposed that combines the amplitude spectroscopy of cable input impedance with neural networks to identify capacitive faults characterized by various degrees of buffer layer ablation and local aging.(3)This paper evaluates the classification performance of neural networks using metrics such as confusion matrices and ROC curves, demonstrating that the Multilayer Perceptron (MLP) neural network can more accurately identify buffer layer ablation faults. Furthermore, experimental validation confirms the effectiveness of the proposed method presented in this paper.

## 2. Cable Fault Model

### 2.1. Cable Structure

A typical high-voltage single-core XLPE cable is generally composed of seven components, arranged from the innermost to the outermost: conductor core, conductor shield, XLPE shield, insulating shield, buffer layer, wrinkle aluminum sheath, and outer sheath, as shown in Figure 1.

#### Mechanism of Buffer Layer Ablation Faults

The chosen research subject is a single-core coaxial cable with a wrinkle aluminum sheath structure, model YJLW 64/110 kV. The two-dimensional axisymmetric model of this cable is constructed in COMSOL Multiphysics 6.0 simulation software, as shown below in Figure 2.

During normal cable operation, both voltage and current are present simultaneously. The current primarily flows axially within the core, generating heat that diffuses radially, causing temperature fluctuations across the cable layers. Alongside the core current, the voltage induces capacitive current within the cable insulation. This capacitive current then travels radially towards the wrinkle aluminum sheath, dissipating into the ground and causing temperature variations across the layers along the pathway.

Following simulation validation, it is determined that the temperature increase induced by the core current in the insulation layer is negligible. Throughout the practical operation of high-voltage cables, the interaction of gravity and diverse operational circumstances leads to irregular contact between the buffer layer and the wrinkle aluminum sheath, potentially resulting in inadequate contact. Consequently, this paper concentrates on examining the impact of different levels of contact between the buffer layer and the wrinkle aluminum sheath on the cable’s capacitive current and temperature elevation.

In the simulation, one valley is set to maintain optimal contact with a 0.1 mm interference fit, while the other valleys along the cable axis intentionally lack contact with the buffer layer, creating a 0.1 mm gap to simulate inadequate contact between the buffer layer and the wrinkle aluminum sheath. The buffer layer is assumed to be damp, and the resulting current density map is depicted in Figure 3 below:

In Figure 3, when only one valley makes contact with the buffer layer, the capacitive current exhibits a symmetrical distribution around the contact point. Only this single contact point allows the current to flow into the wrinkle aluminum sheath and eventually into the ground. Therefore, the current density at points A and B experiences a sharp increase, with a peak value reaching 2.92 × 10^3^ mA/m^2^. The three-dimensional temperature distribution of the cable at this moment is shown in Figure 4 below:

From the above Figure 4, it can be observed that the current density sharply increases at the valley of contact, leading to increased heating at the contact point. The peak temperature is 0.027 °C higher than the ambient temperature, reaching 20.027 °C, and this peak temperature occurs at the midpoint of the buffer layer.

The length of inadequate contact along the cable axis, represented as L, is calculated as the total of the lengths of poor contact on both sides of the valleys with contact. By increasing the length of inadequate contact L and considering the variations in the relative permittivity and resistivity of the buffer layer, the local maximum temperature rise is analyzed under both dry and damp conditions. The simulation results are as follows:

According to Table 1, it is evident that as the length of inadequate contact increases, there is a significant rise in temperature within the buffer layer. Particularly in damp conditions, when the length of inadequate contact reaches 3 m, the temperature rise due to capacitive currents peaks at 329 °C. However, the long-term temperature resistance value of the buffer layer material is only 90 °C, and its instantaneous temperature resistance value is 230 °C, both of which exceed the temperature range of the buffer layer. This will lead to the ablation of the buffer layer and structural damage to the cable.

Therefore, in scenarios where the buffer layer is damp, and there is inadequate contact between the buffer layer and the wrinkle aluminum sheath, once the length of inadequate contact surpasses a certain threshold, the local temperature increase might surpass the melting point of the buffer layer material, resulting in buffer layer ablation faults.

### 2.2. Cable Distributed Parameter Model

According to the transmission line theory, power cables under high-frequency power supply are equivalent to the distributed parameter models, which are used to describe the energy transfer characteristics in the power cable. The equivalent model of the cable distributed parameters is shown in Figure 5.

### 2.3. Cable Local Fault Model

#### 2.3.1. Model of Buffer Layer Ablation Faults

The buffer layer ablation faults imply damage to both the buffer layer and the wrinkle aluminum sheath. As the ablation faults in the buffer layer progress, they can even pose a threat to the integrity of the insulating shield layer. Such circumstances lead to changes in the distributed parameters of the corresponding layers, primarily caused by variations in the material’s properties and geometric structure. Therefore, simulating ablation faults by altering the distributed parameters of a specific section of the cable is feasible, with these adjustments primarily achieved through setting fault factors $\alpha_{1}$, $\alpha_{2}$, and $\beta_{m}$.

At high frequencies, the distributed resistance at the buffer layer ablation faults can be determined by the following equation [20]:$$R_{1}(\omega )\approx \frac{1}{2\pi }\sqrt{\frac{\mu_{0}\omega }{2}}(\frac{1}{r_{c}}\sqrt{\rho_{c}}+\frac{\alpha_{1}}{r_{s}}\sqrt{\rho_{s}})$$
where $R_{1}$ is the distributed resistance of the faulty section; $\alpha_{1}$ is the fault factor of the wrinkle aluminum sheath, which is determined by the fault severity of the wrinkle aluminum sheath, and its value is greater than 1.

The distributed inductance at the buffer layer ablation faults can be determined by the following equation [20]:$$L_{1}(\omega )\approx \frac{\mu_{0}}{2\pi }\ln\frac{r_{s}}{r_{c}}\alpha_{2}+\frac{1}{4\pi }\sqrt{\frac{2\mu_{0}}{\omega }}(\frac{1}{r_{c}}\sqrt{\rho_{c}}+\frac{\alpha_{1}}{r_{s}}\sqrt{\rho_{s}})$$
where $L_{1}$ is the distributed inductance of the faulty section; $\alpha_{2}$ is the mutual inductance fault factor between the conductor core and wrinkle aluminum sheath, and its value is less than 1.

The inner and outer semiconductor layers and the buffer layer have semiconductor characteristics, which have little effect on the magnetic flux inside the cable. Therefore, in the simulation, only the admittance is considered, and its inductance is not considered [21]. For the distributed capacitance and conductance of the cable, the total admittance of multiple dielectric layers can be used for equivalence. The unit length admittance of the dielectric layer *k* can be calculated according to the following formula [22]:$$Y_{k}(\omega )=j\omega \varepsilon_{k}^{\prime }(\omega )\frac{2\pi }{\ln(\frac{r_{k+1}}{r_{k}})}$$
where $Y_{k}(\omega )$ is the unit length admittance of the dielectric layer *k*; $\varepsilon_{k}^{\prime }(\omega )$ is the complex permittivity of the dielectric layer *k*; and $r_{k}$ and $r_{k+1}$ are the outer radii of each dielectric layer.

The dielectric properties of the main insulation material of the cable can be described using the complex permittivity $\varepsilon^{\prime }=\varepsilon +j\omega \sigma$. The expressions for the capacitance *C*_1_ and conductance *G*_1_ per unit length of each layer are [22]:
$$C_{1}(\omega )=\varepsilon (\omega )\frac{2\pi }{\ln(\frac{r_{k+1}}{r_{k}})}\beta_{m}$$
$$G_{1}(\omega )=\sigma (\omega )\frac{2\pi }{\ln(\frac{r_{k+1}}{r_{k}})}\beta_{m}$$
where $\beta_{m}$ is the fault factor of the corresponding layer of the faulty section.

The characteristic parameters of the propagation coefficient γ(ω) and characteristic impedance $Z_{0}$ are, respectively, as follows [20]:$$\gamma (\omega )=\sqrt{\left(R+j\omega L)(G+j\omega C\right)}$$
$$Z_{0}=\sqrt{\left(R+j\omega L)/(G+j\omega C\right)}$$

Buffer layer ablation faults are different from simple local damage. They not only lead to local damage in the cable but also trigger chemical reactions such as moisture absorption and oxidation, resulting in certain changes in local material parameters. Therefore, before and after the faults, it is necessary to adjust not only the resistivity and relative permittivity of the corresponding layers but also to simulate the buffer layer ablation of different severities using fault factors.

The different severity levels of buffer layer ablation faults include mild ablation, where both the wrinkle aluminum sheath and the buffer layer are affected, and severe ablation, where the wrinkle aluminum sheath, the buffer layer, and the insulating shield layer are affected. The fault factors for simulating buffer layer ablation faults are shown in Table 2 [23], where $\beta_{1}$, $\beta_{2}$, and $\beta_{3}$ represent the capacitive fault factors of the wrinkle aluminum sheath, the buffer layer, and the insulating shield, respectively.

#### 2.3.2. Other Fault Models

The aging of cables has a minimal impact on the electrical conductivity and magnetic permeability of the metal layer. Instead, it primarily leads to a decrease in the insulation performance of organic insulation materials, while the relative dielectric constant of the inner and outer semiconductor layers remains essentially unchanged [24]. For the local aging faults of the cable, it belongs to capacitive faults, primarily causing an increase in the distributed capacitance $C_{1}$ of the shield layer. However, the effects on the distributed resistance $R_{1}$, distributed reactance $L_{1}$, and distributed conductance $G_{1}$ of the fault section can be ignored. In order to classify and identify the local faults of the cable in the next chapter, so the local aging faults of the cable are more similar to the frequency domain impedance spectroscopy of the buffer layer ablation faults, a severe aging fault is chosen for the simulation, and is designated as $C_{1}$ = 1.08 $C_{0}$; for inductive faults, the distributed capacitance of the XLPE shield layer is mainly reduced [24], and is designated as $C_{1}$ = 0.98 $C_{0}$.

## 3. Frequency Domain Impedance at the First End of the Cable

### 3.1. Input Impedance of a Normal Cable

Assume the length of the cable is l, and the load impedance connected at the end of the cable is $Z_{L}$. The established model of a normal cable is shown in Figure 6. Taking the cable end as the origin and considering the cable as the x-axis, the positive direction is from the cable’s end towards the cable’s beginning. x represents the distance from the origin to any position. The characteristic impedance of the normal cable is $Z_{0}$, and the propagation coefficient is $\gamma_{0}$.

The reflectance coefficient of the cable [23] is
$$\Gamma_{L}=\frac{Z_{L}-Z_{0}}{Z_{L}+Z_{0}}$$
When the total length of the cable is l, the equivalent input impedance at the first end of the cable [23] is
$$Z_{l}=Z_{0}(\frac{1+\Gamma_{L}\cdot e^{-2\gamma l}}{1-\Gamma_{L}\cdot e^{-2\gamma l}})$$

### 3.2. Input Impedance of Cable with Faulty Section

In a cable, local faults can change the cable’s distributed parameters, resulting in variations in propagation coefficients and characteristic impedance between the faulty section and the normal sections. Thus, based on these differences in characteristic parameters, the cable can be segmented into two normal sections and one faulty section [24]. A scenario of a local fault in the cable is illustrated in Figure 7.

The impedance $Z_{la}$ of the normal section from coordinate $x=l_{a}$ to x=0 is
$$Z_{la}=Z_{0}(\frac{1+\Gamma_{L}\cdot e^{-2\gamma la}}{1-\Gamma_{L}\cdot e^{-2\gamma l_{a}}})$$

$\Gamma_{2}$ is the reflectance coefficient at the end of the faulty section $x=l_{a}$, as follows:$$\Gamma_{2}=\frac{Z_{la}-Z_{1}}{Z_{la}+Z_{1}}$$

The load of the faulty section from coordinates $x=l_{b}$ to $x=l_{a}$ can be equivalent to impedance $Z_{la}$ starting with $l_{a}$; then, the impedance $Z_{lb}$ of the faulty section from coordinates $x=l_{b}$ looking towards $x=l_{a}$ is given by
$$Z_{lb}=Z_{0}(\frac{1+\Gamma_{2}\cdot e^{-2\gamma_{1}(l_{b}-l_{a})}}{1-\Gamma_{2}\cdot e^{-2\gamma_{1}(l_{b}-l_{a})}})$$

$\Gamma_{3}$ is the reflection coefficient at the first of the faulty section $x=l_{b}$, as follows:$$\Gamma_{3}=\frac{Z_{lb}-Z_{0}}{Z_{lb}+Z_{0}}$$

By applying the same approach, the input impedance from the first end of the cable at x=l can be found as follows:$$Z_{l}=Z_{0}(\frac{1+\Gamma_{3}\cdot e^{-2\gamma_{0}(l-l_{b})}}{1-\Gamma_{3}\cdot e^{-2\gamma_{0}(l-l_{b})}})$$

The local fault affects the cable’s input impedance by changing the cable’s distributed parameters. Therefore, based on the established mathematical models for both a normal cable and a cable containing a faulty section, an impedance spectroscopy simulation is conducted on the first end of the cable.

### 3.3. Simulation of Cable Impedance Spectroscopy for Different Fault Types

In the MATLAB 2020b simulation software, simulation models are constructed with different types of faults. The simulation parameters are established as follows: the length of the cable is 100 m, the cable is open-ended, and the frequency range is set from 0 to 100 MHz. The fault is positioned at 50 m along the cable, simulating mild and severe ablation faults of 50 cm in length in the buffer layer, local aging faults of 50 cm and 70 cm in length, along with a 50 cm long inductive fault.

Various types of fault models are created using MATLAB simulation software. The simulation parameters are configured as follows: the cable length is 100 m, with an open circuit at one end, and a frequency range from 0 to 100 MHz. The faults are located 50 m from the first end of the cable, simulating mild ablation faults and severe ablation faults, each 50 cm in length, in the buffer layer. Additionally, local aging faults of 50 cm and 70 cm in length, along with a 50 cm long inductive fault, are simulated.

The simulation selected a 110 kV high-voltage XLPE coaxial cable as the research subject, and the parameters of the constructed cable model [23] are shown in Table 3.

The results of the frequency domain impedance spectroscopy that was performed at the first end of the normal cable through a simulation are illustrated in Figure 8 and Figure 9 below.

The above figures, respectively, show the amplitude spectroscopy and phase spectroscopy of the frequency domain impedance at the first end of the normal cable. They clearly demonstrate that both the amplitude spectroscopy and phase spectroscopy exhibit a certain periodicity, attenuation, and rapid changes. As the frequency gradually increases, the amplitude of the cable’s input impedance shows periodic peaks, aligning with the frequency at which the input impedance’s phase reaches zero. Additionally, both the amplitude and phase of the input impedance exhibit a gradual attenuation trend. Significant changes in the spectroscopy occur around the peaks of the amplitude spectroscopy and the zero-crossing points of the phase spectroscopy.

The input impedance of the cable section with local faults can be iteratively calculated based on the model analysis of the faulty cable section, as described in Section 2.3. Below are the corresponding frequency domain impedance spectroscopy results from the first end of the cable:

From Figure 10, Figure 11 and Figure 12, it can be observed that the frequency domain impedance spectroscopy at the cable’s first end, corresponding to different local faults, exhibits shifts in both the amplitude spectroscopy and phase spectroscopy. However, there is a notable distinction between inductive faults and capacitive faults. When comparing the normal cables, those with inductive faults exhibit a rightward shift, a larger resonance point, and an increase in both the magnitude and phase of the input impedance spectroscopy. Conversely, cables with capacitive faults, including local aging and ablation faults, show a leftward shift, a smaller resonance point, and a decrease in both the magnitude and phase of the input impedance spectroscopy. Therefore, these significant changes in input impedance characteristics can effectively differentiate between inductive and capacitive faults.

Both the buffer layer ablation faults and local aging fault are capacitive faults, causing shifts to the left in both their magnitude spectroscopy and phase spectroscopy. Consequently, solely relying on impedance spectroscopy may make it challenging to accurately differentiate between the buffer layer ablation faults and local aging faults. Therefore, combining frequency domain impedance spectroscopy with artificial intelligence becomes necessary to identify the buffer layer ablation faults.

## 4. Combining Artificial Intelligence for Buffer Layer Ablation Fault Identification

Machine learning, a subfield of artificial intelligence, utilizes computer algorithms to automatically learn patterns and knowledge from data, enabling prediction, decision-making, or inference. Its strength lies in handling complex, nonlinear, high-dimensional data, demonstrating a good generalization ability and robustness. Thus, it can extract and analyze feature information from impedance spectroscopy, establishing fault identification models.

MLP is an artificial neural network based on the perceptron model with a feedforward structure. It can map input variables to output variables, allowing for prediction and modeling. It overcomes the limitations of single-layer perceptron and efficiently handles nonlinear problems. Its network structure is as shown in Figure 13 and mainly consists of three parts: (1) the input layer, consisting of a group of perceptron units or source nodes; (2) the hidden layer, consisting of one or multiple layers of computational nodes; and (3) the output layer, consisting of one layer of computational nodes [25]. Each layer of neurons is a fully connected structure, with the number of neurons in the input and output layers determined by the dimensions of the input and output signals, while the number of hidden layers and the number of hidden nodes in each layer depend on the specific situation.

The signals in each layer of this neural network undergo a transformation through a propagation function and a kernel function, enabling mapping from high-dimensionality to low-dimensionality space. The signals in the output layer further pass through an activation function, ensuring unique output mapping for each input signal [25]. This is represented by the following equation:$$X_{j}^{L}=f(\sum_{i\in M_{j}}x_{i}^{L-1}k_{ij}^{L}+b_{j}^{L})$$
where the overall output layer is represented by $M_{j}$; each input layer has an additional bias b.

### 4.1. Dataset Establishment and Model Building

#### 4.1.1. Dataset Establishment and Preprocessing

Given the random nature of the occurrence of faults in both location and length during the actual operation of high-voltage cables, it is necessary to generate a large number of samples for simulation, training, and testing. For normal cables, 600 sets of samples are configured. For the three types of cables with local faults, a total of 1800 cases are randomly created, involving varying locations and lengths of local faults. A dataset is constructed using the 2400 sets of frequency domain impedance amplitude spectroscopy data of cables obtained through the simulation.

During the process of applying neural network models, raw data typically contain noise and other issues, and the specifications and distributions of the data can vary significantly. Therefore, data preprocessing is essential to enhance the accuracy and stability of the neural network models. In this paper, the Standard Scaler is utilized to standardize the dimensionless data, while a Principal Component Analysis (PCA) [26] is employed to reduce dimensionality and extract primary features from the data.

#### 4.1.2. Establishment and Parameter Settings of Neural Network Models

The MLP neural network model is constructed by using PyTorch (a deep learning framework). In order to compare the results with this model, five other models are also built: Support Vector Machines (SVMs), K-nearest Neighbors (KNN), Logistic Regression (Logistic), Random Forest (RF), and Decision Tree (DT).

The 2400 preprocessed data samples are encoded with values ranging from 1 to 4 to represent the normal cables and three different local faults of the cables, as shown in Table 4 below. And the dataset is divided into a training set and a test set according to the ratio of 4:1; 1920 sets of samples are used for training, and the remaining 480 sets of samples are used for the validation of the classification and identification effect.

To achieve better experimental results, the experiments are repeatedly conducted, in accordance with the previous work of scholars, leading to the determination of the basic parameters of the MLP neural network model, as shown in Table 5.

## 5. Evaluation of Network Results

This paper employs the PyTorch framework to perform comparative experiments on six machine learning classification algorithms: SVM, KNN, Logistic, DT, RF, and the MLP neural network. In these experiments, the number of epochs for all six different classification models is set to 100. The encoding for the classification is set as 1 to 4, respectively, representing the conditions of the cables under normal conditions, mild buffer layer ablation faults, severe buffer layer ablation faults, and local aging faults. The experimental results are outlined below:

From Figure 14, it is evident that all the classification models can accurately identify normal cables with a 100% accuracy rate. However, for cables with severe local aging faults, the SVM model achieves the lowest identification accuracy at 69%. In contrast, the RF, DT, and MLP neural networks achieve an identification rate of 100%, indicating superior performance in recognizing this fault. Yet, significant differences exist in the identification accuracy for the two fault types across the different models: mild ablation faults and severe ablation faults. Among them, the SVM exhibits the poorest identification performance, with only a 48% accuracy rate for severe ablation faults. Conversely, MLP achieves identification accuracy rates of 98% and 97% for the two different severity levels of buffer layer ablation, respectively, showcasing its superior identification performance for these faults.

According to Figure 15, it can be observed that all the classification models exhibit an excellent identification performance in distinguishing between the normal cables and the local aging faults. However, the True Positive Rate (TPR) and AUC values for the DT model are relatively low, with an AUC of only 0.8829, indicating that the algorithm’s performance is not as outstanding as the others. In contrast, the MLP model achieves an AUC of 0.9958, and the TPR values for each class are nearly close to 1, indicating that this model has a higher sensitivity and demonstrates a better classification and identification performance.

The experimental results for the accuracy, precision, and recall evaluation metrics, along with the confusion matrix and ROC curve, are as follows:

Observing Figure 16, it becomes evident that as the number of epochs increases, the classification identification performance of the MLP neural network model steadily improves and stabilizes around 40 epochs. In contrast, the SVM neural network model exhibits an almost opposite trend: as the epochs increase, the classification identification performance of the neural network model weakens and fails to converge even after 100 epochs. When compared to the other five neural network models, the MLP neural network model demonstrates a superior accuracy.

The data from Table 6 clearly show that the MLP neural network model outperforms other models in terms of the accuracy, recall rate, AUC value, and precision. This indicates its superior performance and strong generalization ability.

## 6. Experimental Validation

This paper uses a 100-meter-long 110 kV XLPE coaxial cable as the experimental subject. The cable specimens included a normal cable; a cable with an approximately 60 cm local aging fault located 38 m from the cable end; a cable with mild buffer layer ablation measuring approximately 52 cm at 35 m from the cable end; and a cable with severe buffer layer ablation measuring approximately 43 cm at 40 m from the cable end. Subsequently, impedance spectroscopy analysis is performed using an impedance analyzer at the first end of each cable specimen, and the resulting data are presented in Figure 17 and Figure 18.

After processing the impedance spectroscopy data shown in Figure 17, they are input into the trained MLP neural network for testing. The test results are shown in Table 7 below.

By referring to Table 7, it can be observed that the method accurately identifies the normal cables, the cables with local aging, mild buffer layer ablation, and severe buffer layer ablation. This demonstrates the effectiveness and engineering practicality of the method proposed in this paper.

## 7. Conclusions

In response to the limited methods and low accuracy in identifying cable buffer layer ablation faults, this paper applies artificial intelligence and frequency domain impedance spectroscopy for the identification of cable faults. It proposes a high-voltage cable buffer layer ablation fault identification method based on artificial intelligence and frequency domain impedance spectroscopy. This method demonstrates an excellent capability for extracting fault features and a high accuracy in identifying fault types.
(1)A simulation based on the input impedance model of the cable under both normal and local fault conditions yields the corresponding cable impedance spectroscopies, which are used to differentiate between capacitive and inductive faults. Additionally, preprocessing is applied to the magnitude spectroscopies of the input impedance, and the preprocessed data are used as the input samples for the MLP neural network to train the network.(2)The simulation results show that the cable buffer layer ablation fault identification method, utilizing MLP neural networks and frequency domain impedance spectroscopy, can effectively identify various faults including mild and severe ablation, local aging, and normal cables, achieving an identification accuracy of 97.5%. Additionally, the effectiveness of this method is verified by experiments.

## Figures and Tables

**Figure 1 sensors-24-03067-f001:**
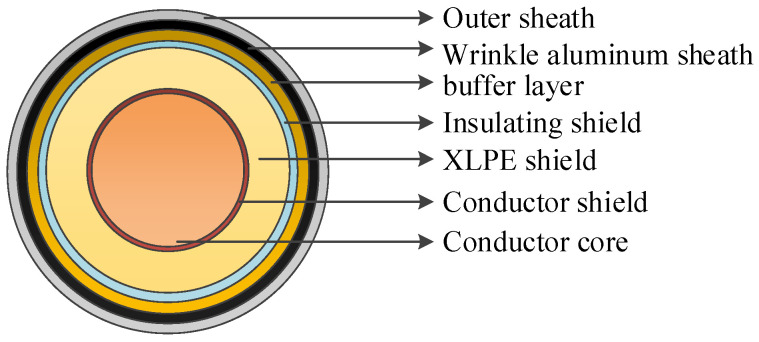
High-voltage single-core cable structure diagram.

**Figure 2 sensors-24-03067-f002:**
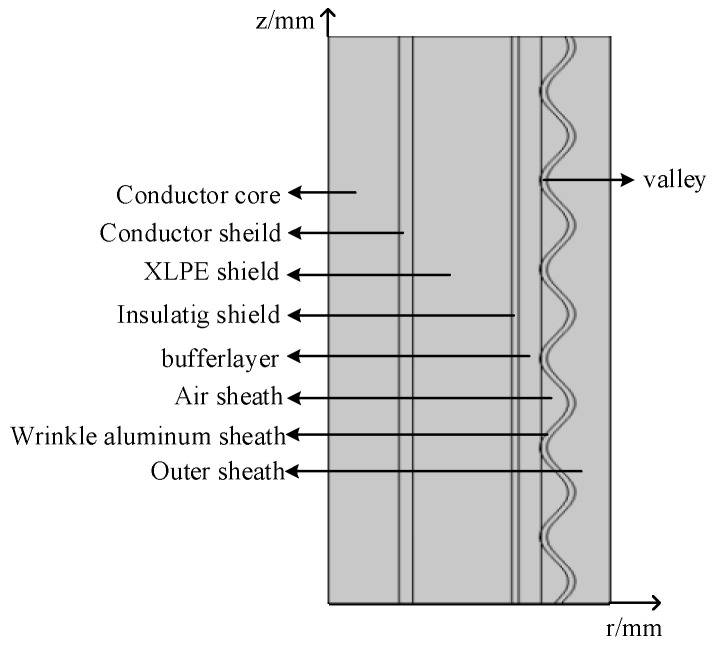
Axial profile of cable.

**Figure 3 sensors-24-03067-f003:**
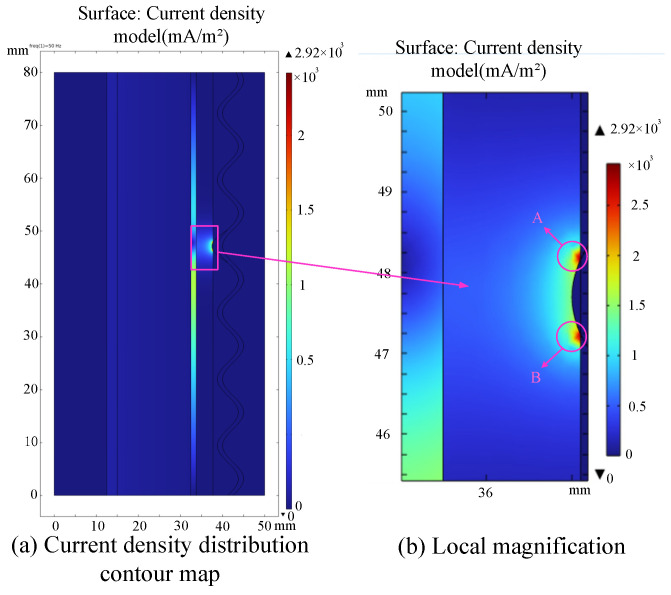
The current density distribution of the cable under inadequate contact.

**Figure 4 sensors-24-03067-f004:**
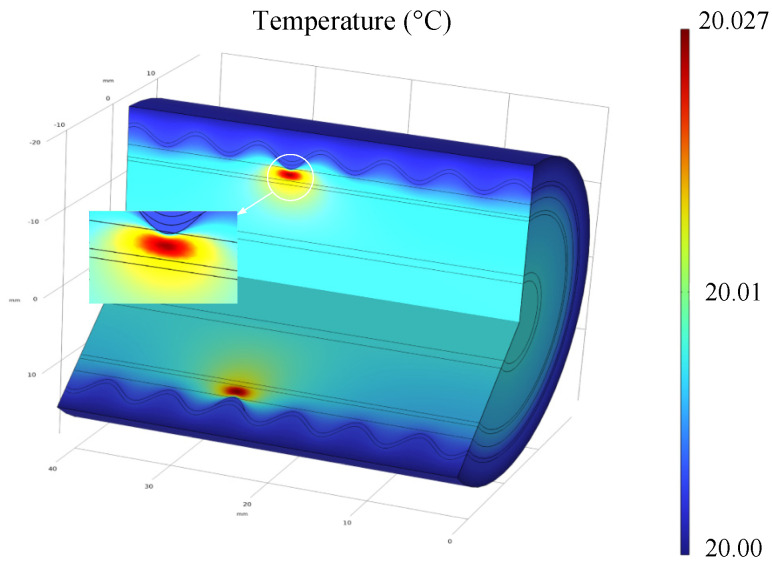
The temperature distribution diagram of the cable under inadequate contact.

**Figure 5 sensors-24-03067-f005:**
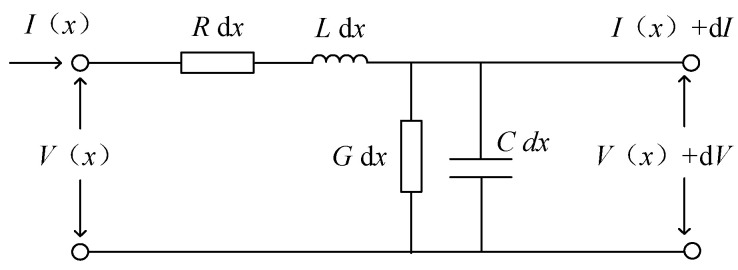
Schematic diagram of cable shaft cross-section.

**Figure 6 sensors-24-03067-f006:**
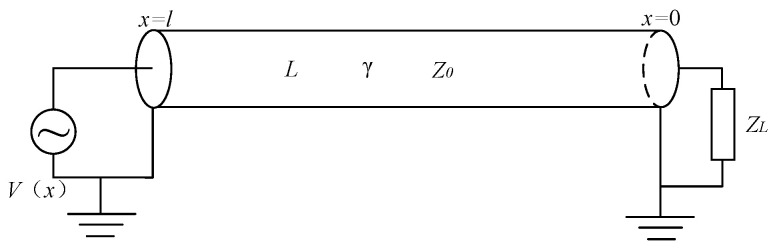
Normal cable transmission model.

**Figure 7 sensors-24-03067-f007:**
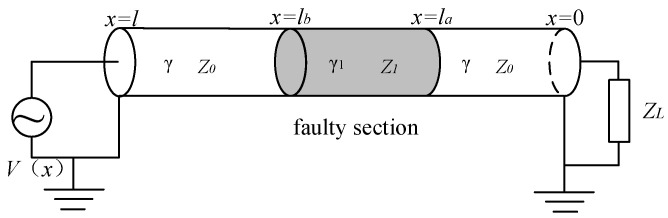
Cable transmission model with a faulty section.

**Figure 8 sensors-24-03067-f008:**
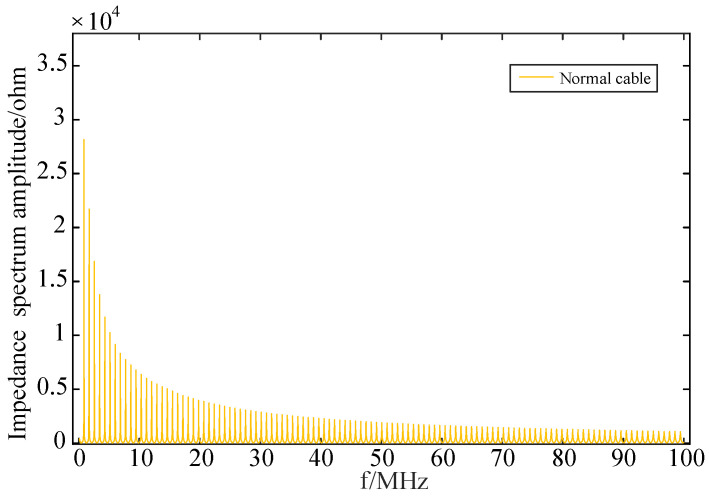
Frequency domain impedance amplitude spectroscopy of normal cable.

**Figure 9 sensors-24-03067-f009:**
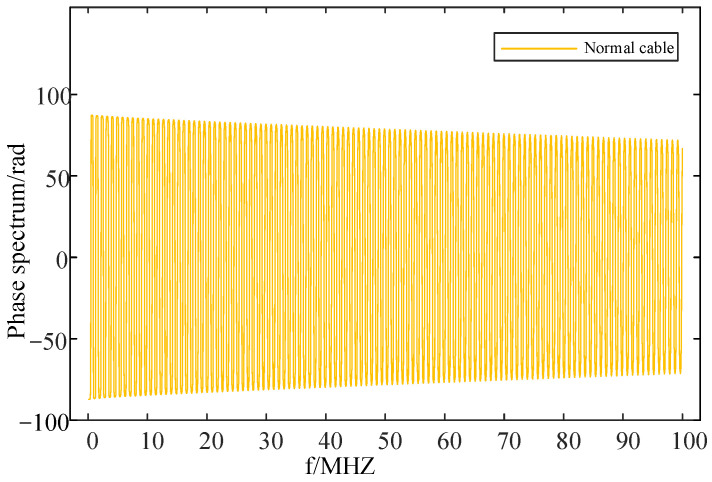
Frequency domain impedance phase spectroscopy of normal cable.

**Figure 10 sensors-24-03067-f010:**
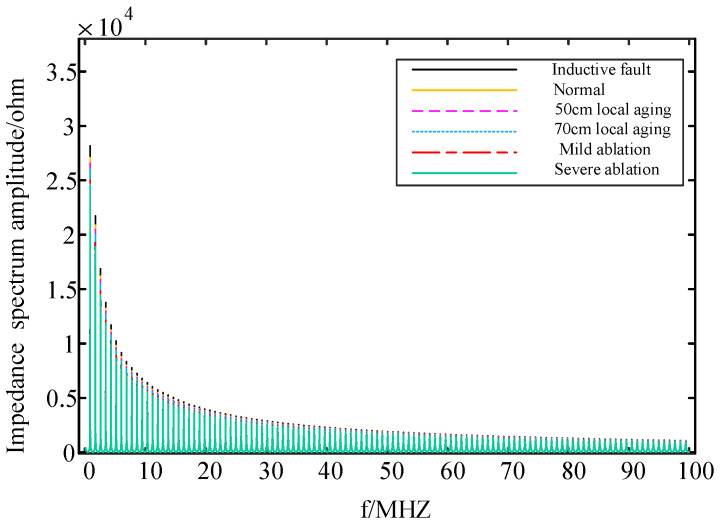
Amplitude spectroscopy of input impedance at the first end of the cable corresponding to different faults.

**Figure 11 sensors-24-03067-f011:**
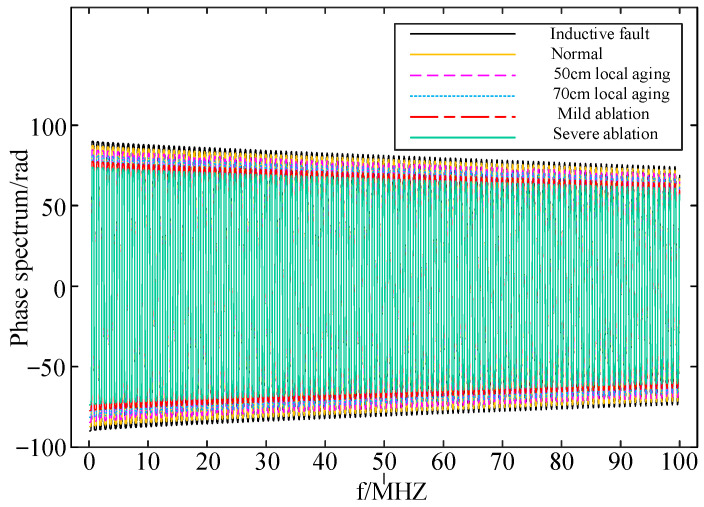
Phase spectroscopy of input impedance at the first end of the cable corresponding to different faults.

**Figure 12 sensors-24-03067-f012:**
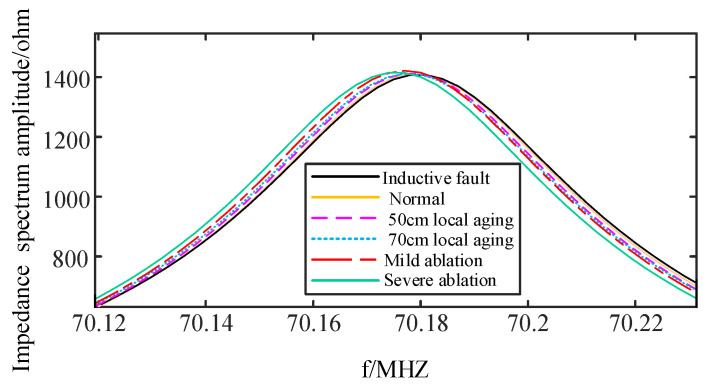
Local magnification of impedance spectroscopy corresponding to different faults.

**Figure 13 sensors-24-03067-f013:**
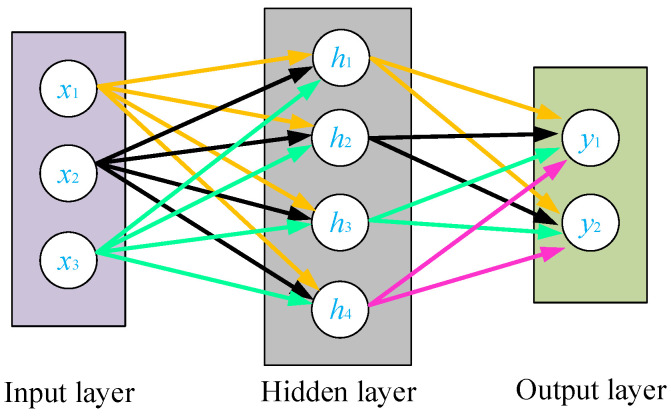
Multilayer perceptron (MLP) neural network structure diagram.

**Figure 14 sensors-24-03067-f014:**
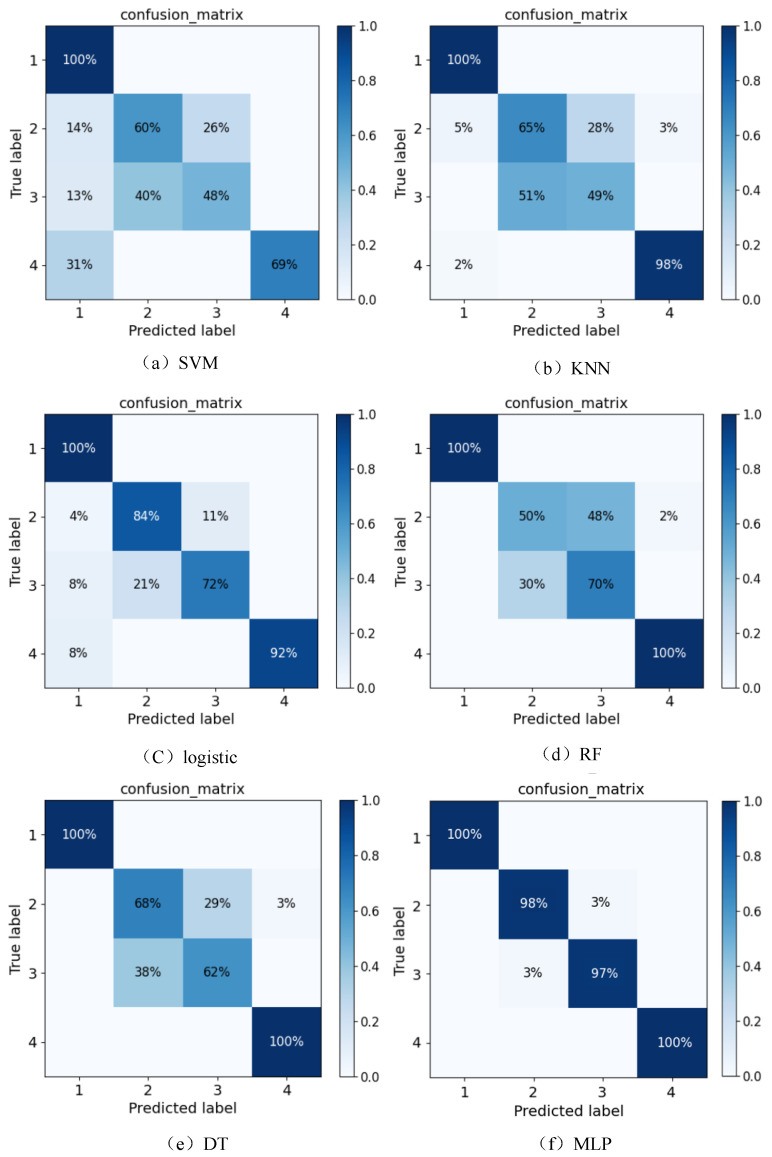
Classification result confusion matrix diagram.

**Figure 15 sensors-24-03067-f015:**
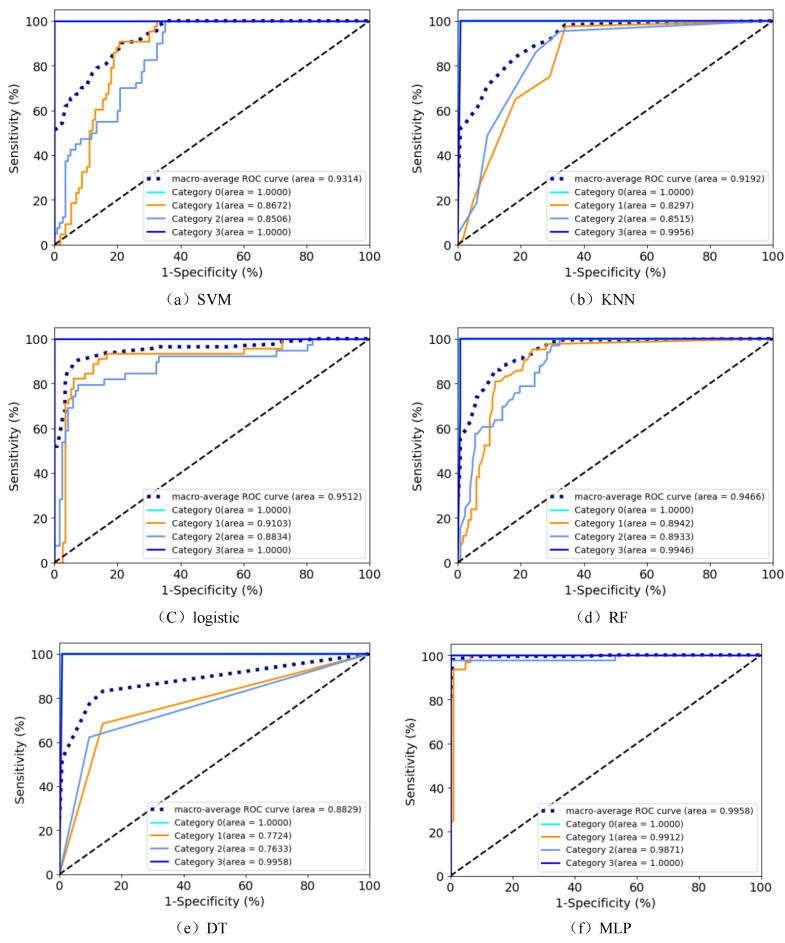
ROC curve comparison chart.

**Figure 16 sensors-24-03067-f016:**
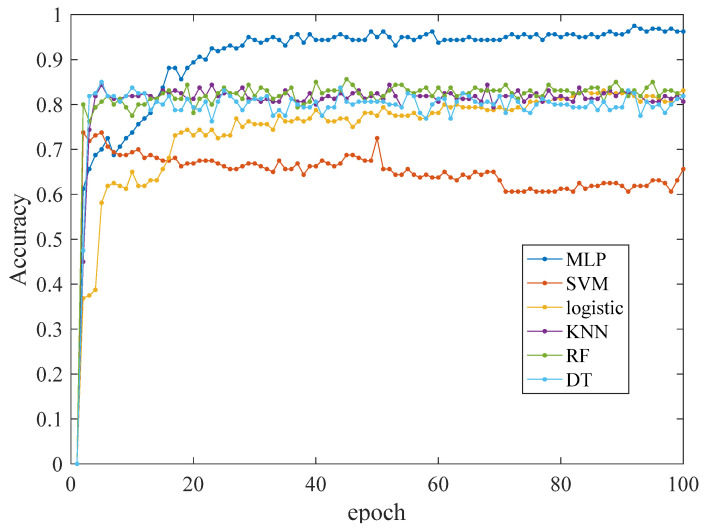
Comparison of identification accuracy between six model test sets.

**Figure 17 sensors-24-03067-f017:**
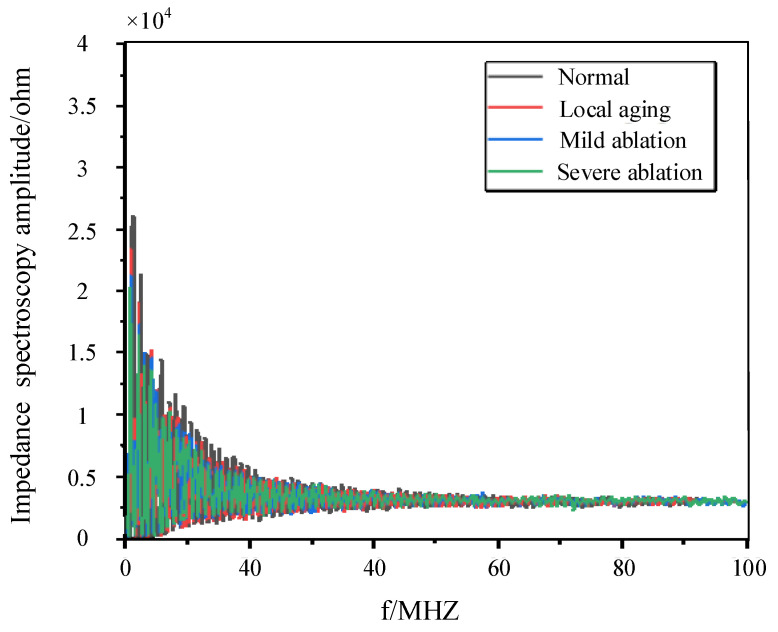
The magnitude spectroscopy of the cable input impedance.

**Figure 18 sensors-24-03067-f018:**
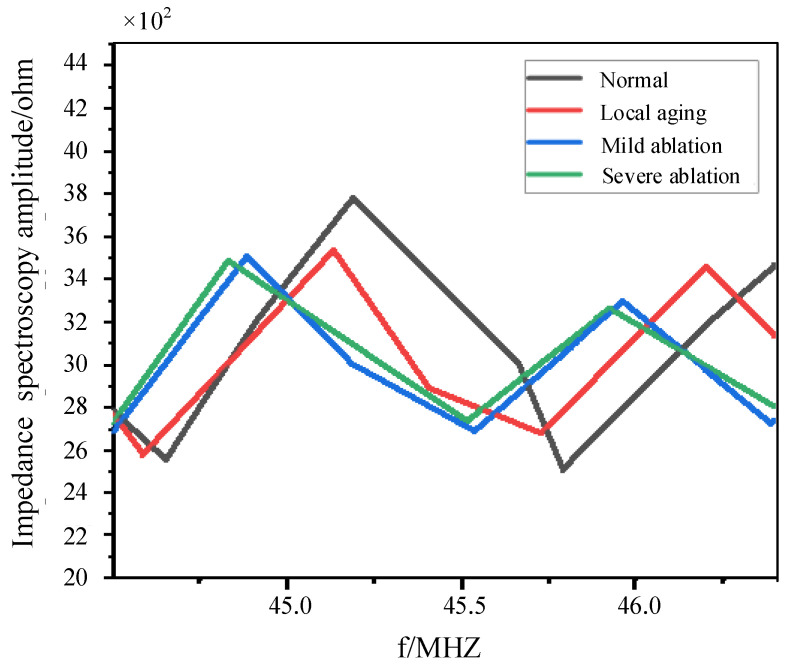
Local magnitude spectroscopy of the cable input impedance.

**Table 1 sensors-24-03067-t001:** The simulation results of the relationship between the length of inadequate contact (L) and the temperature rise (T) of the buffer layer.

Length of Inadequate Contact L/m	Temperature Rise When the Buffer Layer Is Dry/°C	Temperature Rise When the Buffer Layer Is Damp/°C
0.25	0.067	6.04
0.4	0.106	12.95
0.55	0.215	19.16
0.70	0.323	27.38
1.0	0.534	48.02
3.0	3.2566	329.05

**Table 2 sensors-24-03067-t002:** The fault factors for simulating buffer layer ablation faults.

Severity Levels of Fault	*α* _1_	*α* _2_	*β* _1_	*β* _2_	*β* _3_
Mild ablation fault	2	0.8	0.85	0.85	1
Severe ablation fault	2	0.6	0.75	0.75	0.85
Without fault (normal)	1	1	1	1	1

**Table 3 sensors-24-03067-t003:** YJLW 64/110 kV cable structure parameters.

Structure	Inner Diameter/mm	Outside Diameter/mm	Relative Permittivity	Resistivity/(Ω/(kg·K))
Conductor core	0	31.9	1	1.8 × 10^−6^
Conductor shield	31.9	34.4	700	5
XLPE shield	34.4	67.4	2.3	1 × 10^16^
Insulating shield	67.4	69.4	700	5
Buffer layer	69.4	84.5	100	1 × 10^5^
Corrugated aluminum sheath	84.5	95.5	1	2.8 × 10^−6^

**Table 4 sensors-24-03067-t004:** Coding of different fault types in cables.

Type	The Encoding for Different Faults
Normal	1
Mild ablation fault	2
Severe ablation fault	3
Local aging	4

**Table 5 sensors-24-03067-t005:** Basic parameters of MLP model.

Parameter	Value
Learning rate	0.001
Enter the number of layer neurons	The number of principal components after dimensionality reduction
Number of hidden layers	2
Number of neurons in hidden layer 1	8
Number of neurons in hidden layer 2	4
Hidden layer activation functions	Sigmoid
The number of neurons in the output layer	4
Batch size	1
Epoch	100
Loss function	Cross-entropy loss function

**Table 6 sensors-24-03067-t006:** Six model identification results.

Model	Acc	Recall	AUC	Precision
SVM	0.6563	0.5974	0.9314	0.6662
KNN	0.65	0.6123	0.9192	07089
Logistic	0.80	0.7845	0.9512	0.8277
DT	0.8187	0.8057	0.8829	0.8178
RF	0.8189	0.7968	0.9466	0.8272
MLP	0.975	0.98	0.9958	0.98

**Table 7 sensors-24-03067-t007:** Results of the proposed method.

Type	Probability of Normal	Probability of Local Aging	Probability of Mild Ablation	Probability of Severe Ablation	Classification Result	Right or Wrong
Normal	0.9838	0.0058	0.0076	0.0028	1	√
Mild ablation	6.5901 × 10^−7^	9.9944 × 10^−1^	5.5764 × 10^−4^	1.9116 × 10^−8^	2	√
Severe ablation	1.0948 × 10^−10^	2.5226 × 10^−3^	9.9748 × 10^−1^	3.7039 × 10^−9^	3	√
Local aging	1.1135 × 10^−5^	1.0443 × 10^−6^	3.5592 × 10^−7^	9.9999 × 10^−1^	4	√

## Data Availability

Data are contained within the article.

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
