# Peer review of "High-Voltage Cable Buffer Layer Ablation Fault Identification Based on Artificial Intelligence and Frequency Domain Impedance Spectroscopy"

_sensors, 2024, doi:10.3390/s24103067_

Round 1

Reviewer 1 Report

Comments and Suggestions for Authors

1.       The text mentions: Compared to a normal cable, the impedance spectrum of an inductive fault will shift to the right, while local aging and buffer layer erosion will shift to the left. However, in actual field testing, it might not be possible to obtain the impedance spectrum information of a normal cable under that model. In such cases, how can one differentiate between inductive and capacitive faults?

2.       In the Matlab simulation mentioned in the text, the lengths of the locally aged sections are set to 50cm and 70cm, and the length of the buffer layer erosion section is set to 50cm. In reality, the lengths of local defect sections may be shorter. In such cases, how do the impedance spectra of different types of defects differ? Is the information contained in the impedance spectra sufficient for classification by neural network algorithms?

3.       Are there any literature references for the specific parameter settings used in the Matlab simulation for simulating buffer layer erosion as shown in Table 1? How is the degree of erosion defined as severe or minor? The data settings in Table 1 seem rather arbitrary.

4.       When using a neural network model, if the randomly created cases only involve different defect positions and lengths, should the settings of parameters related to specific defect types (such as those in Table 1) also be randomly generated?

5.       The encoding for different fault types in Table 3 is 03, while in Figure 11, the encoding for the classification result matrix is 14, which can easily cause confusion. It is recommended to standardize the encoding.

6.       In actual detection, a cable may contain multiple defect sections with different types of defects and varying degrees of severity. How does the neural network model classify in such situations? The model proposed in the article seems to be capable of classifying cables containing only a single type of fault.

7.       The method of combining artificial intelligence with impedance spectroscopy for cable fault identification has already been proposed. Is it convincing to train the model solely with simulated, randomly generated data? It is suggested that the dataset used for model training should be obtained from impedance spectroscopy tests of on-site cables to further enhance the feasibility of field application.

Comments on the Quality of English Language

ensure logical coherence in discussion

Reviewer 2 Report

Comments and Suggestions for Authors

This paper proposes a method for recognizing buffer layer ablation fault based on frequency domain impedance spectroscopy and artificial intelligence.

The authors are suggested to focus on the following points:

1.- The method proposed in this paper is based on frequency domain impedance spectroscopy. I understand that the cable must be disconnected from the installation in order to apply this technique, that is, it cannot be applied on-line. Please provide more details and acknowledge this limitation if it is the case.

2.- Buffer layer ablation fault has not been defined. It would be nice to see photographs or schematics of this fault mode.

3.- Please ensure that all equations are either referred or demonstrated.

4.- Figs. 5-8-9. It shows the Z-f graph, with values of Z in the order of several kOhm. Is it possible such value? At first glance it makes no sense. Please elaborate.

5.- I cannot find experimental results. Simulation results are based on many approximations that can lead to results very far from the expected in real situations.

6.- Machine learning is applied to simulation data, so the results can be much better than those expected in real application based on experimental data.

7.- Please include experimental results, without them the paper cannot be accepted for publication since simulation results can present important differences with respect to experimental results.

8.- The use of gray literature must be avoided.

I hope my comments can help to enhance the quality of a future resubmission of a completely redone manuscript.

Comments on the Quality of English Language

I suggest proofreading the paper carefully.

Round 2

Reviewer 2 Report

Comments and Suggestions for Authors

The authors have replied my concerns and have included experimental data. Please improve the quality of Figure 17.

Comments on the Quality of English Language

The authors are suggested to proofread the manuscript
